# Fracture fixation in the hand and wrist: A 16-year population-based study of 56 163 patients from the Swedish National Patient Register

**Viktor Schmidt**[1,2,3¤]*, **Elsa Pihl**[1,2,3], **Cecilia Mellstrand Navarro**[1,2,3,4], **Michael Axenhus**[1,2,3]

**1** Danderyd Hand and Wrist Initiative, Danderyd Hospital, Stockholm, Sweden, **2** Department of Orthopaedic Surgery, Danderyd Hospital, Stockholm, Sweden, **3** Department of Clinical Sciences at Danderyd Hospital, Karolinska Institutet, Stockholm, Sweden, **4** Department of Clinical Science and Education Södersjukhuset, Karolinska Institutet, Stockholm, Sweden

¤ **Postal address:** Orthopaedic clinic, Entrévägen 2 182 68, Danderyd University Hospital, Danderyd, Sweden.

* viktor.schmidt@ki.se

## Abstract

Hand and wrist fractures are among the most common orthopaedic injuries, with a growing trend toward surgical treatment. However, large-scale data on regional variations and treatment trends remain limited. This population-based study analyses the incidence, trends, and regional variations in hand and wrist fracture fixation with plates and screws in Sweden from 2008 to 2023, with predictive modelling for future trends. A total of 56,163 patients aged ≥15 years underwent fixation (code NDJ69). Southern regions, including Skåne and Halland, had the highest fixation rates (>100/100,000), while northern areas like Norrbotten had significantly lower rates (<20/100,000). Women ≥65 years had the highest incidence. Predictive models indicate a continued increase in procedures, particularly among women aged ≥65, through 2035. These findings highlight regional disparities and the ongoing shift towards surgical treatment in older populations, emphasizing the need for optimized treatment strategies to ensure equitable access to care.

## Introduction

Wrist fractures are the most common fractures in adults [1,2] and their incidence is increasing worldwide [2,3]. Together with hand fractures, they account for up to 30% of emergency department visits [3–5]. Treatment decisions are based on both radiological findings and patient-specific factors. However, there is ongoing debate about the optimal type of surgery and the ideal timing [6–8]. Over the 21st century, surgical intervention has become increasingly preferred [9]. In displaced fractures, surgery more reliably restored anatomical alignment compared with nonsurgical treatment [10,11], which may be crucial for achieving favourable clinical outcomes [12]. Furthermore, early primary surgery appears to yield better hand function compared to delayed primary surgery [7,10].

**Data availability statement:** The data used in this study is obtained from the website of the SNBHW and is publicly available for anyone to download and use. The data sets can be obtained from the NPR directly (https://www.socialstyrelsen.se/en/statistics-and-data/statistics/statistical-databases/).

**Funding:** The author(s) received no specific funding for this work.

**Competing interests:** The authors have declared that no competing interests exist.

Because definitions of acceptable radiological alignment vary, treatment recommendations differ both between and within countries [13–15].

Using large-scale data from the Swedish National Patient Register (NPR), this study aims to analyse demographic variations in incidence rates of fracture fixation—emphasising age and sex differences—evaluate regional disparities to identify high- and low-incidence of surgery areas; and forecast future trends to support healthcare planning.

## Methods

### Study design and setting

This population-based observational study utilizes open-access surgical data obtained from the NPR for the period between 2008 and 2023. The study complies with the RECORD guidelines [16]. The data used in this study is obtained from the website of the SNBHW and is publicly available for anyone to download and use.

### Healthcare system overview

Sweden's National Health Service provides universal healthcare access to all residents, covering emergency care, general hospital services, and outpatient visits at no cost. Although private hospitals exist, fracture cases are universally treated in public facilities. Each resident is assigned a unique Swedish personal identification number that remains valid throughout their lifetime or until emigration [17]. This identifier is integral for interactions within both public and private healthcare systems and is linked to all national healthcare registers.

### Data source

The NPR is a comprehensive registry documenting healthcare data for patients treated within Sweden's closed healthcare system [18]. The NPR has recorded inpatient care since 1964 (nationwide since 1987) and included specialized outpatient services starting in 2001. Updates were annual until 2021 but have been conducted monthly since June 2021, incorporating delayed or corrected data. The registry provides detailed records of surgical procedures, including geographic distribution, age, and sex of patients. All healthcare providers, both public and private, are required to report diagnoses and procedures. This includes ICD-10 diagnoses [19], (since 1994) and surgical procedure codes classified under the NOMESCO framework [20]. Data from all hand surgery and orthopaedic departments in Sweden are included in the NPR. The registry has been reported to provide data of high quality [21,22].

### Patient selection

The study considered individuals aged 15 years or older who underwent fracture fixation with a plate in the hand or wrist, as recorded in the NPR between January 1, 2008, and December 31, 2023. Only individuals with Swedish personal identification numbers were included.

## Study population

Inclusion criteria:

1. Individuals with residency in Sweden at one point during 1st of January 2008 and 31st of December 2023.

2. Individuals who were registered with a fracture fixation in the hand or wrist (NOMESCO code NDJ69).

Exclusion criteria:

1. Age <15 years.

## Statistical analysis

The data was extracted as incidence of surgical procedure per 100,000 inhabitants. Data were stratified by sex, age group, and region to facilitate trend comparisons over time. Incidence rates were controlled by a separate incidence calculation that was done by dividing the number of patients by the total population, as reported by Statistics Sweden [23]. Significant differences between sex in each age groups were assessed using the Student's t-test. To predict future trends, regression analysis was employed, and we fitted exponential, linear, logarithmic, polynomial, and power regression models for each incidence trend. The year was used as the independent variable and incidence of fracture surgery was as the exposure. The assumptions underlying regression analysis were assessed to ensure model validity. Linearity was evaluated visually using scatter plots of residuals versus fitted values. Independence of residuals was checked using the Durbin-Watson statistic. Full stationarity testing not applicable due to the short time series. Model validation was performed by comparing the adjusted $R^2$ across models to ensure generalizability without overfitting. The regression analysis used only data from 2014 to 2023 to account for significant changes in treatment practice which were implemented before 2014. Predictive analysis was based on the best-fitting model, with 95% confidence intervals (CI) applied where relevant. A p-value of <0.05 was considered statistically significant. * indicate <0,05, **<0,01, **<0,001 and ***<0,0001. Regional maps graphs were created in Illustrator 2025 (29.5.1).

## Ethical considerations

This study relied exclusively on open-access data (which is anonymized and aggregated to avoid small-cell risks) and was thus exempt from ethical review as Swedish law does not require ethical permit for the study of group level open access data.

## Results

A total of 56 163 patients were identified with the code NDJ69 during the study period. The overall incidence of fracture fixation increased steadily from 2008 to 2023. The annual incidence trends for fracture fixation with plates and screws from 2008 to 2023 show an increase in fracture fixation for women after 2020 compared to total amount of fracture diagnoses (Fig 1).

Age and sex distributions revealed distinct patterns. The incidence was highest in women aged ≥65 years. Younger patients, 15–64 years, showed relatively stable rates over the study period without notable differences between sexes (Fig 2).

Significant geographic disparities in procedure rates were noted. In 2023, significant geographic disparities in the metric were observed across Sweden. Northern regions like Västernorrlands län (95.7) and Jämtlands län (66.9) had notably high values, while southern areas such as Blekinge län (23.4) and Kalmar län (26.1) were considerably lower. Central regions like Stockholms län (45.4) aligned closely with the national average of 49.9. (Fig 3).

Model fitting found a polynomial regression model of the 2nd order to be best fit (S1 Table). Regression modelling indicated a continued rise in procedure incidence for women while men are expected to have a similar or slightly decreasing trend up until 2035 (Fig 4).

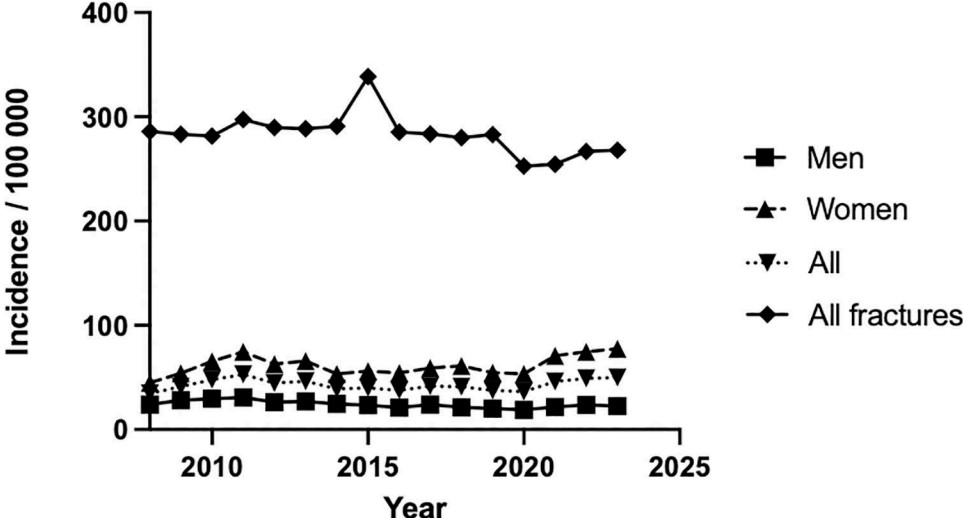

**Fig 1. Trends in hand and wrist fracture fixation with plates and screws, 2008–2023. "All fractures" represent those treated with all modalities (non-operative included).** The other three trend lines (Men, Women and All) represent only fixation using plates and screws.

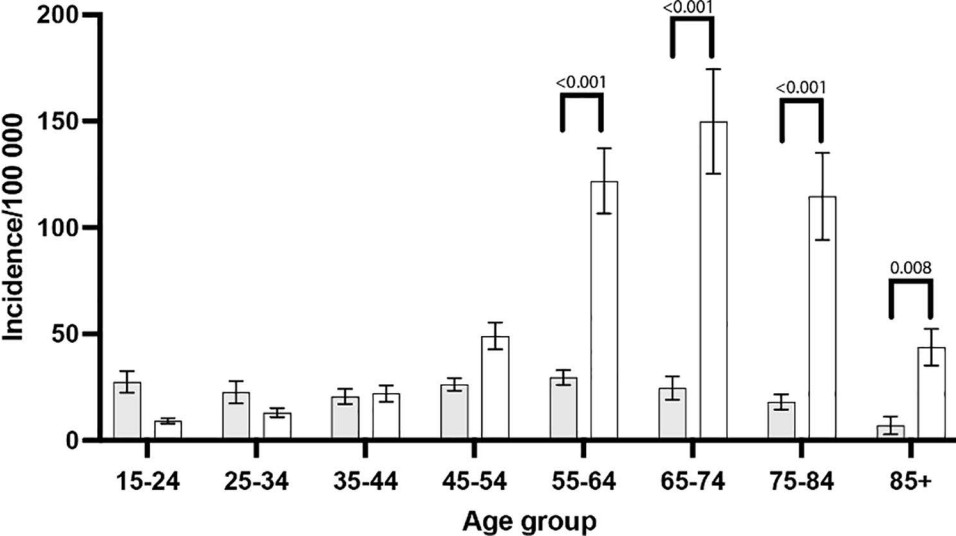

**Fig 2. Age groups of patients who underwent fixation with plate for wrist or hand fractures.** White indicates women, grey indicate men.

## Discussion

Our study reveals a significant increase in the utilization of plate and screw fixation for hand and wrist fractures in Sweden from 2008 to 2023, with a notable rise among women aged 65 and older. Regional disparities were observed, with southern regions like Skåne and Halland exhibiting higher incidence rates compared to northern areas such as Norrbotten. Predictive models suggest a continued annual increase of these procedures amongst women through 2035.

The increasing utilization of plate and screw fixation for hand and wrist fractures in Sweden reflects a global trend toward favouring internal fixation methods [9]. This shift is particularly evident among women aged ≥65, a demographic

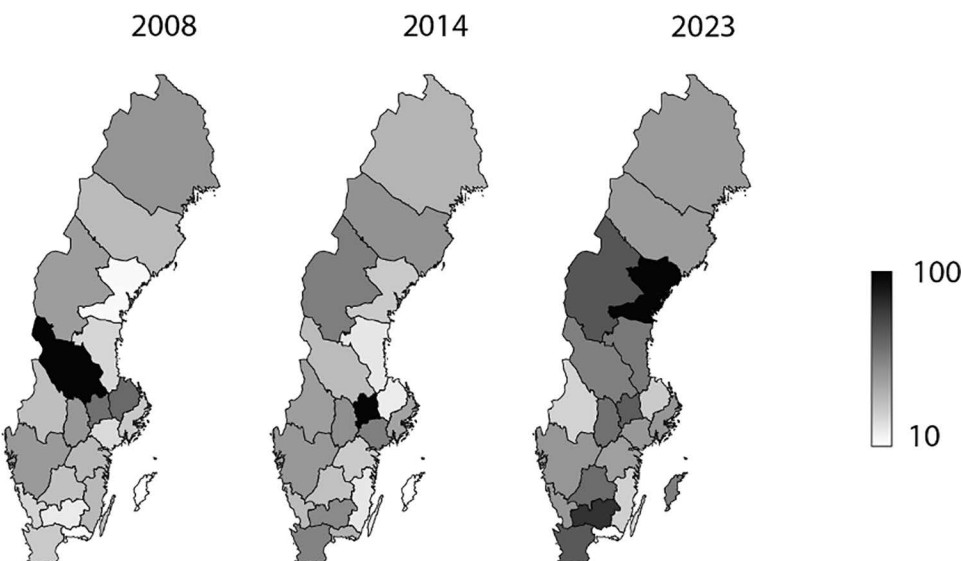

**Fig 3. Regional variance in fracture fixation in the hand and wrist using plate and screws, comparison between 2008, 2014 and 2023.** Incidence is marked as a colour gradient from 10 to 100 per 100,000 inhabitants.

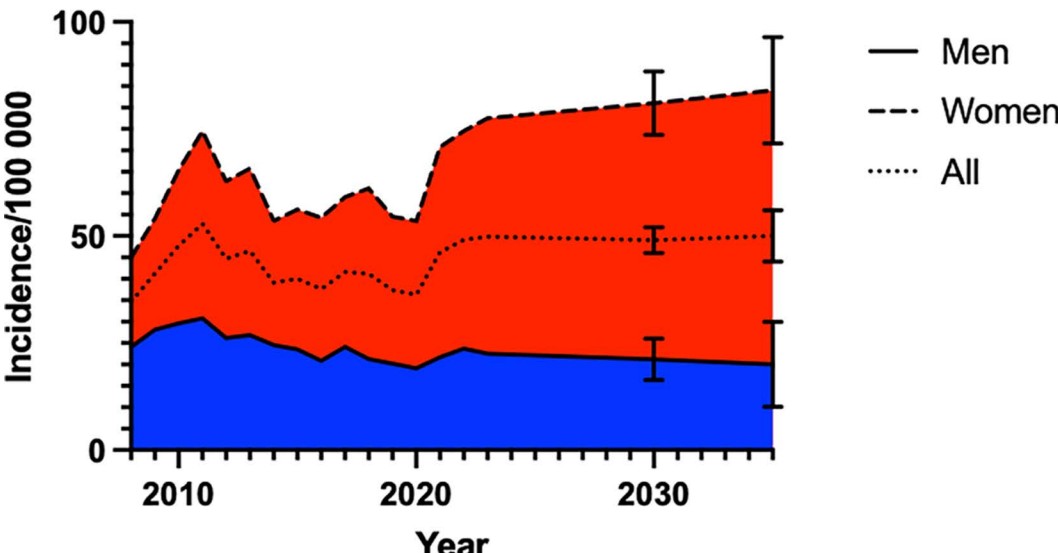

**Fig 4. Future trend predication of hand and wrist fracture fixation.**

with a higher prevalence of osteoporosis, making them more susceptible to fractures [24]. The implementation of Sweden's national guidelines in 2021, advocating for prompt surgical intervention for specific unstable distal radius fractures, seem to have significantly influenced clinical practice [25,26]. These guidelines recommend surgery within one week of injury to enhance patient outcomes amongst patients with high functional demands. The national guidelines are also partly radiologically based with certain criteria prompting surgery. Our study period, 2008–2023, encompasses the introduction of these guidelines, and while a direct causal relationship cannot be conclusively established, the observed increase in

surgical interventions during this time suggests a correlation. Moreover, such guidelines might draw attention to surgical treatment of vital elderly patients who otherwise might have been subject to a conservative regime due solely to their chronological age, increasing equity along with the operative incidence, this might in turn lead to improved clinical outcomes [8,26].

The national guidelines aim to standardize treatment protocols across various patient demographics, ensuring equitable care and minimising unwarranted treatment variations. This standardization is expected to lead to more predictable outcomes and improved overall patient satisfaction. Implementing these guidelines necessitates adequate resources, including trained personnel and surgical facilities, to accommodate the increased demand for timely surgical interventions. However, the effectiveness of these guidelines may vary depending on the level of implementation at different hospitals [26]. Healthcare systems must ensure that resources are equitably distributed to prevent regional disparities. Educating patients about the benefits and risks associated with early surgical intervention is crucial. Shared decision-making should be encouraged, considering individual patient preferences, risks, and expected outcomes. National quality registries play a pivotal role in collecting data to evaluate the effectiveness of implemented guidelines and inform future revisions. Due to increasing volume of surgery, implant removal might become a future concern for healthcare providers while surgeries addressing complications such as correctional osteotomies might decrease.

Regional variations in procedure rates are notable, with southern regions such as Blekinge and Kalmar showing higher incidences compared to northern regions like Norrbotten. Several factors may contribute to these disparities. Southern Sweden's higher population density and greater access to specialized orthopaedic services may facilitate more frequent surgical interventions. In addition, regional differences in medical tradition and adherence to national guidelines can influence treatment choices. Variations in age distribution, increasing number of older patients and the prevalence of osteoporosis across regions may also contribute to differing fracture rates and treatment decisions. These findings point to broader differences in healthcare delivery and utilization that warrant further study. In particular, the potential benefits of increased use of plate fixation amongst women is yet to be determined.

Research suggests that surgical variation primarily stems from differences in physician beliefs about the indications for surgery, as well as the degree to which patient preferences shape treatment decisions [27,28]. These factors, in turn, are influenced by the environmental variables discussed above.

The increasing adoption of plate and screw fixation for hand and wrist fractures in Sweden has significant clinical implications. The aim of early fixation is to enable favourable conditions for early mobilization, which is crucial in preventing joint stiffness and promoting functional recovery. However, the rise in these procedures necessitates that orthopaedic surgeons maintain proficiency in advanced fixation techniques as well as proper patient selection to ensure optimal outcomes. Although the incidence of surgical fixation has increased in Sweden, it remains unclear if this has led to increased patient satisfaction and clinical outcomes. Whether the implementation of a nation-wide guidelines has had positive healthcare effects remains unanswered and is an area for further study. It must also be mentioned that definitive treatment remains a clinical decision in which surgeons must considering factors like bone quality and fracture type, to determine the most suitable treatment approach.

## Limitations

The accuracy of the NPR depends on consistent and precise reporting from healthcare providers. Variations in coding practices, potential miscoding, transcription errors, underreporting, and missing data may influence data quality, as is typical in register-based studies. These variations could differ between regions. The specific procedural code NDJ69 represents open reduction and internal fixation of a fracture in the wrist or hand with plates and screws. It does not include other modalities or anatomical sites. Additionally, the observational nature of our study limits our ability to control for confounding factors such as patient comorbidities, socioeconomic status, and bone quality, which may influence treatment decisions and outcomes. The NPR primarily records procedural data and lacks detailed clinical outcomes, such as

functional recovery and patient satisfaction, restricting our ability to assess the long-term effectiveness of the surgical interventions studied. Furthermore, the study does not account for patients treated non-surgically or with alternative methods, introducing potential selection bias. Future research should consider incorporating comprehensive patient information, employing advanced statistical methods to adjust for potential confounders, and conducting prospective studies to establish causal relationships and evaluate long-term outcomes.

## Conclusion

Our study reveals a significant increase in plate and screw fixation for hand and wrist fractures in Sweden from 2008 to 2023, particularly among women aged 65 and older. Southern regions like Blekinge and Kalmar exhibit higher incidence rates compared to northern areas such as Norrbotten. Predictive models indicate an annual increase in these procedures amongst women through 2035. The implementation of national guidelines in 2021, advocating for prompt surgical intervention for distal radius fractures, appears to have influenced these trends. Further research is warranted in order to determine if increased surgical fixation correlate to patient satisfaction and clinical outcomes.

## Supporting information

**S1 Table. Adjusted R2 values for tested regression models.**
(DOCX)

## Author contributions

**Conceptualization:** Viktor Schmidt, Cecilia Mellstrand Navarro, Elsa Pihl, Michael Axenhus.

**Data curation:** Michael Axenhus.

**Formal analysis:** Michael Axenhus.

**Investigation:** Michael Axenhus.

**Methodology:** Viktor Schmidt, Michael Axenhus.

**Project administration:** Viktor Schmidt, Michael Axenhus.

**Resources:** Michael Axenhus.

**Software:** Michael Axenhus.

**Supervision:** Cecilia Mellstrand Navarro, Elsa Pihl, Michael Axenhus.

**Validation:** Michael Axenhus.

**Visualization:** Michael Axenhus.

**Writing – original draft:** Viktor Schmidt, Michael Axenhus.

**Writing – review & editing:** Viktor Schmidt, Cecilia Mellstrand Navarro, Elsa Pihl, Michael Axenhus.

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
