## [Decision Letter · Decision Letter 0]

20 May 2025

PONE-D-25-13377Fracture fixation in the hand and wrist: A 16-year population-based study of 56 163 patients from the Swedish National Patient RegisterPLOS ONE

Dear Dr. Schmidt,

Thank you for submitting your manuscript to PLOS ONE. After careful consideration, we feel that it has merit but does not fully meet PLOS ONE’s publication criteria as it currently stands. Therefore, we invite you to submit a revised version of the manuscript that addresses the points raised during the review process.

ACADEMIC EDITOR: The manuscript was well written but needs some improvement. Please address the reviewer's concerns.

We look forward to receiving your revised manuscript.

Kind regards,

Ken Iseri

Academic Editor

PLOS ONE

Journal Requirements:

3. We note that Figure 3 n your submission contain [map/satellite] images which may be copyrighted. All PLOS content is published under the Creative Commons Attribution License (CC BY 4.0), which means that the manuscript, images, and Supporting Information files will be freely available online, and any third party is permitted to access, download, copy, distribute, and use these materials in any way, even commercially, with proper attribution. For these reasons, we cannot publish previously copyrighted maps or satellite images created using proprietary data, such as Google software (Google Maps, Street View, and Earth). For more information, see our copyright guidelines: http://journals.plos.org/plosone/s/licenses-and-copyright.

1. You may seek permission from the original copyright holder of Figure 3 to publish the content specifically under the CC BY 4.0 license. 

Additional Editor Comments:

The manuscript was well written but needs some improvement. Please address the reviewer's concerns.

Reviewers' comments:

Reviewer's Responses to Questions

**Comments to the Author**

1. Is the manuscript technically sound, and do the data support the conclusions?

Reviewer #1: Yes

Reviewer #2: Yes

2. Has the statistical analysis been performed appropriately and rigorously? 

Reviewer #1: Yes

Reviewer #2: Yes

3. Have the authors made all data underlying the findings in their manuscript fully available?

Reviewer #1: Yes

Reviewer #2: Yes

4. Is the manuscript presented in an intelligible fashion and written in standard English?

Reviewer #1: Yes

Reviewer #2: Yes

5. Review Comments to the Author

Reviewer #1: Thank you for the opportunity to review this study.

There is a minor typo on Page 10, Line 243. The word "to" should come before "enable".

I have just one other suggestion for the authors to consider that was prompted by their comments in the Limitations section of the paper, noting that bone quality may influence treatment decisions and outcomes. I have read that the Swedish National Patient Register (NPR) does not include data on prescription medications. This information is held on the Swedish Prescribed Drug Register (SPDR) as the authors will know. A brief literature search identified several linkage studies combining data from both registries such as a 2020 study on surgical site infections after distal radius fracture surgery (see https://bmcmusculoskeletdisord.biomedcentral.com/articles/10.1186/s12891-020-03822-0) and a 2021 study that explored associations between hormonal contraception and antidepressant use (see https://bmjopen.bmj.com/content/11/10/e049553).

It would be interesting to see an NPR-SPDR linkage study to evaluate trends in prescription of osteoporosis specific medications among patients undergoing fracture fixation of the wrist, or indeed other types of fragility fractures occurring among people aged 50 years and older.

In the event that the authors have any intentions in the future to undertake linkage studies with data from the NPR and SPDR, readers of their publication may be interested to know of this future direction.

Reviewer #2: This is a valuable, well-structured, and timely manuscript that analyzes long-term national data on hand and wrist fracture fixation procedures in Sweden. The use of a large, population-based dataset over a 16-year span provides robust epidemiological insights, particularly regarding temporal trends, regional variation, and demographic characteristics. The topic is highly relevant for both clinical orthopaedics and healthcare planning, especially in light of aging populations and increasing surgical intervention rates.

The manuscript is generally well-written, the data are clearly presented, and the conclusions are supported by the results. However, several areas require further clarification, elaboration, and refinement. Below I outline specific strengths and areas for improvement.

1.Major Strengths

Comprehensive National Dataset: The use of the Swedish National Patient Register (NPR) ensures a high degree of population representativeness and statistical power. The inclusion of 56,163 patients provides a strong foundation for trend analysis.

Public Health Relevance: The findings have clear implications for health resource allocation, surgical training, and guideline implementation. The observed increase in fixation procedures among elderly women and regional disparities are important from a health equity perspective.

(3)Use of Predictive Modelling: The attempt to forecast future surgical trends through regression modelling adds value, particularly for anticipating future healthcare demands.

2.Major Points for Revision

①. Methodological Transparency of Predictive Modelling

While multiple regression models (linear, exponential, etc.) were tested, the manuscript does not provide sufficient detail on model selection criteria. The reporting of R² values is not enough—please include a table comparing AIC/BIC or adjusted R² values for each model type to support your choice of best-fitting models.Additionally, assumptions behind the regression analysis (e.g., linearity, independence, stationarity) should be briefly discussed, and model validation (if any) should be mentioned.

②. Data Limitations and Potential Biases

The study relies on procedural coding (NDJ69), but it is unclear whether this code uniformly represents only open reduction and internal fixation with plates and screws. Could this include other modalities or anatomical sites beyond the wrist/hand? Please clarify the specificity of this code.The manuscript mentions possible regional differences in clinical practices. However, another likely contributor is variation in data reporting or coding fidelity across regions. Consider discussing this potential source of bias and its impact on results.

③. Interpretation of Regional Disparities

While geographic variations are noted, explanations remain somewhat speculative. A deeper discussion of possible factors—such as socioeconomic differences, distribution of orthopedic specialists, hospital capacity, or urban–rural demographics—would significantly enhance the interpretive strength.Consider supplementing Figure 3 with quantitative data (e.g., in a supplementary table) for transparency and replicability.

④. Impact of National Guidelines (2021)

The paper discusses the introduction of Swedish national guidelines in 2021, but does not quantitatively analyze their impact. Given the timeline (2008–2023), this represents an opportunity for an interrupted time series (ITS) analysis or stratification of data before/after 2021 to demonstrate policy impact more convincingly.

⑤Language and Style

Overall, the manuscript is well-written, though some sections would benefit from stylistic tightening.Ensure consistent use of English spelling conventions (e.g., “favourable” vs. “favorable”).The phrase “elderly women” appears frequently—consider using more neutral phrasing such as “women aged 65 years and older” in some contexts.

⑦Figures

Figure 1 caption should clearly define “All fractures” and clarify if it includes non-surgical management.Color schemes in Figure 3 could be made more colorblind-friendly and readable.

Please indicate the statistical significance of trends (p-values or confidence intervals) directly in the figures where appropriate.

⑧Ethical Considerations

Although the study uses publicly available data and is exempt from ethical approval, a brief statement on data anonymization and privacy safeguards would be helpful.

6. PLOS authors have the option to publish the peer review history of their article (what does this mean? ). If published, this will include your full peer review and any attached files.

**Do you want your identity to be public for this peer review?** For information about this choice, including consent withdrawal, please see our Privacy Policy .

Reviewer #1: No

Reviewer #2: No

---

## [Author Response · Author response to Decision Letter 1]

10 Jul 2025

1. Please note that PLOS ONE is unable to publish previously copyrighted maps or satellite images, or images created using proprietary data. For these reasons, we cannot publish images generated by software which copyrights their output (such as Google Maps, Street View, and Earth). In order to use these images in your submission, we require explicit permission from the copyright owner to publish the figures under the CC BY 4.0 license.

At this time, please kindly clarify the following regarding Figure 3:

a) Where did the authors obtain the maps in Figure 3?

We have drawn the map ourselves in Illustrator.

b) Please state whether the map have been previously copyrighted to your knowledge.

The outlines of the map is our own and we have also used it in similar images in other PLOS ONE publications.

c) If any of the map images have been previously copyrighted,

No.

---

## [Decision Letter · Decision Letter 1]

27 Jul 2025

Fracture fixation in the hand and wrist: A 16-year population-based study of 56 163 patients from the Swedish National Patient Register

PONE-D-25-13377R1

Dear Dr. Schmidt,

We’re pleased to inform you that your manuscript has been judged scientifically suitable for publication and will be formally accepted for publication once it meets all outstanding technical requirements.

Kind regards,

Ken Iseri

Academic Editor

PLOS ONE

Additional Editor Comments (optional):

Reviewers' comments:

Reviewer's Responses to Questions

**Comments to the Author**

1. If the authors have adequately addressed your comments raised in a previous round of review and you feel that this manuscript is now acceptable for publication, you may indicate that here to bypass the “Comments to the Author” section, enter your conflict of interest statement in the “Confidential to Editor” section, and submit your "Accept" recommendation.

Reviewer #2: All comments have been addressed

2. Is the manuscript technically sound, and do the data support the conclusions?

Reviewer #2: Yes

3. Has the statistical analysis been performed appropriately and rigorously? 

Reviewer #2: Yes

4. Have the authors made all data underlying the findings in their manuscript fully available?

Reviewer #2: No

5. Is the manuscript presented in an intelligible fashion and written in standard English?

Reviewer #2: Yes

6. Review Comments to the Author

Reviewer #2: (No Response)

7. PLOS authors have the option to publish the peer review history of their article (what does this mean? ). If published, this will include your full peer review and any attached files.

**Do you want your identity to be public for this peer review?** For information about this choice, including consent withdrawal, please see our Privacy Policy .

Reviewer #2: No

---

## [Editor Report · Acceptance letter]

PONE-D-25-13377R1

PLOS ONE

Dear Dr. Schmidt,

I'm pleased to inform you that your manuscript has been deemed suitable for publication in PLOS ONE. Congratulations! Your manuscript is now being handed over to our production team.

Kind regards,

on behalf of

Dr. Ken Iseri

Academic Editor

PLOS ONE